# Dynamic Activity Index for Feature Engineering of Geodynamic Data for Safe Underground Isolation of High-Level Radioactive Waste

**Sergei M. Agayan [1], Ilya V. Losev [1,2,\*], Ivan O. Belov [1], Victor N. Tatarinov [1,3] , Alexander I. Manevich [1,2] and Maksim A. Pasishnichenko [1]**

[1] Geophysical Center RAS, Molodezhnaya Str., 3, 119296 Moscow, Russia; s.agayan@gcras.ru (S.M.A.); i.belov@gcras.ru (I.O.B.); v.tatarinov@gcras.ru (V.N.T.); a.manevich@gcras.ru (A.I.M.); m.pasishnichenko@gcras.ru (M.A.P.)

[2] Mining Institute of NUST «MISiS», Leninsky Prospekt 4, Building 1, 119049 Moscow, Russia

[3] Russia Institute of Physics of the Earth, O.Yu. Schmidt RAS, Moscow, B. Gruzinskaya Str., 10, Building 1, 123242 Moscow, Russia

\* Correspondence: i.losev@gcras.ru

**Abstract:** In this study, we developed a new approach for feature engineering in geosciences. The main focus of this study was feature engineering based on the implementation of the dynamic activity index (MDAI) as a function of the anomaly of the spatial distribution of data, using systems and discrete mathematical analysis. The methodology for calculating MDAI by groups, geomorphological variability, the density of tectonic faults, stress-strain state, and magnetic field anomalies, is presented herein for a specific area. A detailed analysis of the correlation matrix of MDAI revealed weak correlations between the development features. This showed that the considered properties of the geological environment are independent sets and can be used in the analysis of its geodynamic stability. As a result, it was found that most of the territory where high-level radioactive waste (HLRW) disposal is currently planned is in a relatively stable zone.

**Keywords:** system analysis; DMA algorithms; dynamic activity index; structural tectonic block; geodynamic data; safety

## 1. Introduction

Methods of discrete mathematical analysis, machine learning, and big data analysis use "feature" in their terminology. The feature of a study object is the result of measuring or modeling some individual property of the object, and their aggregate reflects the model of the object. The solution to urgent problems in assessing natural and man-made risks, such as searching for anomalies in geophysical fields [1–3], recognizing strong earthquake-prone areas [4–8], geodynamic zoning [9,10], etc., requires the creation of effective methods for the formalized analysis of a complex of geological and geophysical features. When analyzing the data, the features are synthesized using mathematical modeling methods [11–13] and may contain complex mathematical constructions. It is difficult to interpret them physically. In this case, the informativeness of geological and geophysical features is assessed [14–16]. Therefore, the feature model of the object under study should be adequately analyzed, and the results of this analysis should correctly reflect the real natural-technical system.

A unique project is being implemented in Russia to create an underground research laboratory (URL). The main goal of this project is to confirm the possibility of final isolation of high-level radioactive waste (HLRW) in geological formations. Considering international experience, the concept of deep geological repository (DGR) was chosen, the main parameters of which are given in [17]. DGR is planned to be created in granite gneiss rocks of the Nizhnekansky massif 4–5 km from the Yenisei River at a depth of 500–600 m.

HLRW will be placed in large-diameter wells drilled from horizontal workings on horizons with proposed dimensions of $1.0 \times 1.5$ km. The multibarrier protection system consists of classical elements used in various DGR projects (Sweden, Finland, Japan, USA, etc.): geological environment, bentonite, borosilicate glass, and container body. Currently, the geodynamic and seismic regime is being studied in this area [18–20].

The geological and tectonic features of the DGR location area are described in [20–24]. The time of the radiobiological hazard of HLRW exceeds ten thousand years, so the main focus is to assess and predict the geodynamic stability of a structural tectonic block containing HLRW. A structural tectonic block is a dynamically active system in which internal and external energy sources determine the spatiotemporal evolution of its structural forms and stress-strain state. The activity degree of the system is reflected in the features (morphology) of the distribution of geological and geophysical characteristics, including the relief of the Earth's surface, the scheme of tectonic faults and stress-strain state, etc. When such information does not provide an obvious sign of instability [21], system analysis is the most constructive assessment. This allows long-term forecasts in conditions of information uncertainty, uses fundamental geological patterns as the only criteria, and guarantees reliability [25].

For this purpose, and based on the methods and algorithms of discrete mathematical analysis (DMA) and fuzzy logic, a method of formalized analysis was developed [26,27]. It considers the relationship of geodynamics and the morphological features of the distribution of geological and geophysical parameters (including a digital relief model, stress-strain state, geophysical fields, and other characteristics of the environment) [1,26,28,29].

The calculation of dynamic indicators' activity measurements is presented in Section 2. The results of correlation analysis and the final measure of geodynamic safety are presented in Section 3. Assessment of the stability of the geological environment is detailed in Section 4. Finally, in Section 5, conclusions are summarized.

## 2. Methodology for Assessing the Integral Index of Dynamic Stability of a Structural Tectonic Block

Information about the methodology that was developed by the Geophysical Center RAS is detailed below. Its algorithms are based on the language of fuzzy sets and fuzzy logic [26,30]. DMA algorithms have successfully proven themselves in solving a wide range of geological and geophysical problems in the field of Earth sciences (strong recognition of earthquake-prone areas, monitoring of volcanoes and geomagnetic activity, etc.) [26,31–34].

In [26], a finite system of functions of geological and geophysical parameters was studied on a two-dimensional grid. The challenge of calculating the anomaly measurements for a group of geological and geophysical features was formalized. The mathematical component of the sustainability assessment methodology for the structural-tectonic block is described in detail in [1,26] and tested at the HLRW disposal site. Its main provisions are as follows.

The area of interest $\Pi = \{o \leq x_1 \leq t; u \leq x_2 \leq v\}$ exists on the coordinate plane $\mathbb{R}^2(x_1, x_2)$. A set of geological and geophysical fields $F$ from $m$ datasets (in the form of digital maps of various parameters: geographical, geological, geophysical, geodynamic, economic, and others) is selected for this area for use in evaluating the stability of the region $\Pi$.

$$F = \{f_1, \ldots, f_m\},$$
$$f_i : \Pi \to \mathbb{R}; \ i = 1, \ldots, m \tag{1}$$

The goal is to divide the region $\Pi$ into relatively unstable (conditionally dangerous) and stable (conditionally safe) elements. This ranking is also called geodynamic zoning [10].

In the region $\Pi$ a regular grid $W = W(h_1, h_2)$ with nodes $w$ is given:

$$W = \left\{ w = (o + ih_1; u + jh_2) \ \middle| \ \begin{array}{l} i = 0, \ldots, N; \ h_1 = \frac{t-o}{N} \\ j = 0, \ldots, M; \ h_2 = \frac{v-u}{M} \end{array} \right\} \tag{2}$$

On the grid $W$ it is necessary to analyze the spatial distribution of systems of functions $F$ in the neighborhoods of the node $w$. To do this, we defined a fuzzy measure of activity $\mu(w)$ in the range from 0 to 1, according to the rules described below.

Step 1. Calculation of the dynamic indicator.

Each parameter $f$ from the set $F$ is a distribution function on the grid $W$. For each parameter $f$, it is possible to determine the dynamic indicator $D_f$, which is a functional characteristic of the measurement of $f$. The value of $D_f(w)$ is interpreted as a quantitative assessment of the behavior of the function $f$ in the node $w \in W$, calculated according to the specified rules. In terms of data analysis, the dynamic indicator $D_f(w)$ is a feature.

Step 2. Calculation of measure of dynamic activity index (MDAI).

For each dynamic indicator $D_f$, a measure of activity (anomaly) $\mu D_f$ is determined in the range from 0 to 1. It shows the degree of expression of the property $f$ in the node $w$, defined by the indicator $D_f$. The measure of dynamic activity index $\mu D_f$ is calculated from the dynamic indicator $D_f$ in the methodology of discrete mathematical analysis. The transformation $D_f \to \mu D_f$ translates the analysis of the measurement of $f$ into the language of fuzzy logic: measures of dynamic activity index $\mu D_f$ for different dynamic indicators of $D_f$ are fuzzy structures on the grid $W$, and they can be combined in any compositions and any quantities using fuzzy logic operations and averaging.

Step 3. Calculation the integral measure of activity $\mu_F$.

As the last step of the algorithm, all the measures of dynamic activity index $\mu D_f$ are combined into a single integral indicator of $\mu_F$. The formula of the combination is the arithmetic mean of all measures of dynamic activity index $\mu D_f$. Depending on the research task, weighting coefficients for activity measures or other compound formulas can be used [35]. To show the measures of geodynamic safety, the inverse of the integral measure of activity is used:

$$S_F = 1 - \mu_F \qquad (3)$$

The transformation $F \to \mu_F$ translates vector analysis concerning the system of functions $F$ into scalar analysis for the final measure of anomaly $\mu_F$, which, in terms of decision theory, is the reduction of a multicriteria problem to a scalar choice of a utility function. In terms of geodynamic zoning [36], the criteria for estimating the value of $S_F$ according to the system of features $F$ are determined according to Table 1. This approach to the ranking is adequate; if an integral measure is expressed in the range 0–1, then the conditions $S_F \leq 0.25$ и $S_F \geq 0.75$ mean the absence or the presence of safety (anomalies), respectively. The interval $(0.25;\ 0.75)$ indicates uncertainty and the need for additional research.

**Table 1.** Ranking of the integral measure of geodynamic safety $S_F$.

| Node (Cell, Structural Block) $w$ | Measures of Geodynamic Safety $S_F$ |
| :---: | :---: |
| hazardous | $\leq 0.25$ |
| neutral | $\in (0.25;\ 0.75)$ |
| safe | $\geq 0.75$ |

## 3. Feature Engineering Based on Measurement of Dynamic Stability Index

### 3.1. Needed Data

According to the methodology described in [26], when selecting parameters and related dynamic indexes, we used a set of parameters necessary for assessing geodynamic stability. All the initial data were collected in a GIS project [37]. We used a set of the following data:

1. Digital terrain model based on radar interferometric survey of the Earth's surface, Shuttle radar topographical mission (SRTM-4);
2. Schemes of tectonic faults [38,39];
3. Scheme of neotectonic structures of the joint zone of the Siberian platform and the West Siberian plate [40];

4. Kinematic model of modern horizontal movements and rates of deformation of the Earth's crust according to GNSS monitoring data [19,41];
5. Data on the stress-strain state of the Nizhnekansky massif obtained as a result of finite element modeling [42,43];
6. Map of the anomalous magnetic field [38].

Dynamic indexes were obtained from the above datasets based on the study of the following characteristics: relief, faults, stress-strain state, and anomalous magnetic field.

### 3.2. Feature Calculation

For the realization of the methodology, software modules based on the calculation of morphometric indicators and normalization of geological and geophysical parameters were developed. These parameters reflect patterns of relief $L_{Re}^1(w)$, $L_{Re}^2(w)$, and $|\nabla_{Re}|(w)$; fault densities $P(w,\mathcal{P})$ and $P(w, \rho)$; stress-strain states $\sigma_{xx}(w)$, $\sigma_{yy}(w)$, $\sigma_{int}(w)$, $E_{xx}(w)$ $E_{yy}(w)$, and $dil(w)$; and magnetic field anomalies $L_{Mag}^2(w)$, $|\nabla_{Mag}|(w)$.

The first two indicators, $L_{Re}^1$ and $L_{Re}^2$, characterize the geomorphological variability landform sections at node $w$, and the third, $|\nabla_{Re}|$, is the gradient of relief:

$$L_{Re}^1(w) = \frac{\sum_{\overline{w} \in C(w)} |Re(\overline{w}) - Re(w)|}{4} \tag{4}$$

$$L_{Re}^2(w) = \frac{2 + cos\theta_\Gamma + cos\theta_B}{2} \tag{5}$$

$$cos\theta_\Gamma = \frac{-1 + (Re(w_4) - Re(w)) \times (Re(w_6) - Re(w))}{\sqrt{1 + (Re(w_4) - Re(w))^2} \times \sqrt{1 + (Re(w_6) - Re(w))^2}} \tag{6}$$

$$cos\theta_B = \frac{-1 + (Re(w_2) - Re(w)) \times (Re(w_8) - Re(w))}{\sqrt{1 + (Re(w_2) - Re(w))^2} \times \sqrt{1 + (Re(w_8) - Re(w))^2}} \tag{7}$$

The third relief gradient index is the gradient modulus $|\nabla_{Re}(w)|$; it is responsible for the maximum change in relief at a node $w$ and calculated by the Sobel operator:

$$|\nabla_{Re}(w)| = \left|\nabla_{Re}^\Gamma(w)\right| + \left|\nabla_{Re}^B(w)\right| \tag{8}$$

$$\nabla_{Re}^\Gamma(w) = (Re(w_7) + 2Re(w_8) + Re(w_9)) - (Re(w_1) + 2Re(w_2) + Re(w_3)) \tag{9}$$

$$\nabla_{Re}^B(w) = (Re(w_3) + 2Re(w_6) + Re(w_9)) - (Re(w_1) + 2Re(w_4) + Re(w_7)) \tag{10}$$

The MDAI for $L_{Re}^1$, $L_{Re}^2$, and $|\nabla_{Re}|$ are calculated as:

$$\begin{aligned}
\mu L_{Re}^1(w) &= \frac{L_{Re}^1(w)}{L_{Re}^1(w) + \overline{L_{Re}^1}} \\
\mu L_{Re}^2(w) &= \frac{L_{Re}^2(w)}{L_{Re}^2(w) + \overline{L_{Re}^2}} \\
\mu \nabla_{Re}(w) &= \frac{\nabla_{Re}(w)}{\nabla_{Re}(w) + \overline{\nabla_{Re}}}
\end{aligned} \tag{11}$$

In studying the stability of a structural-tectonic block, the key issue is the macroscopic manifestation of the geodynamic process. In the Earth's crust, this macroscopic manifestation is most often represented by the relative displacement of parts of a massif concerning each other. Analysis of the processes occurring at the boundaries between structural-tectonic blocks is a necessary step for the creation of modern analytical and numerical geodynamic models [44]. For this, we added the indicators $P(w)$, which characterizes fault density, and $P(\text{п},\mathcal{P})$, which characterizes proximity to tectonic faults.

The MDAI of fault density was determined using the linear density, which was obtained in a circular vicinity within each cell of the grid. The length of the segment of each line crossed by the circular neighborhood was multiplied by the line weight factor. Then all the length values were summed up and divided by the area of the circle. This process was

repeated for all cells in the grid. The calculation of the values $\rho(\Pi,\mathcal{P})$ from a cell $\Pi$ to the fault $\mathcal{P}$ was obtained using the Kolmogorov mean criterion with a negative index:

$$\rho(\Pi,\ \mathcal{P}) = \begin{cases} 0, </mo> \in \mathcal{P} \\ M_q\left(\rho(\Pi,\ P_k)|_1^N\right) \end{cases}, \tag{12}$$

where

$$q < 0$$

$$M_q\left(\rho(\Pi,\ P_k)|_1^N\right) = \left(\frac{\sum_{k=1}^N \rho(\Pi,\ P_k)^q}{N}\right)^{\frac{1}{q}} \tag{13}$$

The MDAIs of $\rho(\Pi,\mathcal{P})$ are calculated using the formula:

$$\mu\rho(,\ \mathcal{P}) = \rho(,\ \mathcal{P})\rho(,\ \mathcal{P}) + \rho(,\ \mathcal{P}) \tag{14}$$

Figure 1 shows the original digital elevation model of the HLRW disposal area. Figures 2–4 show the calculated maps of elevation variability for $L^1_{Re}(w)$, $L^2_{Re}(w)$, and $|\nabla_{Re}|(w)$, respectively. An example of the calculation of the proximity to tectonic faults $\rho(w,\mathcal{P})$ as the shortest distance to a tectonic fault is given in Figure 5; an example of the measure of fault density $\rho(w)$ is given in Figure 6.

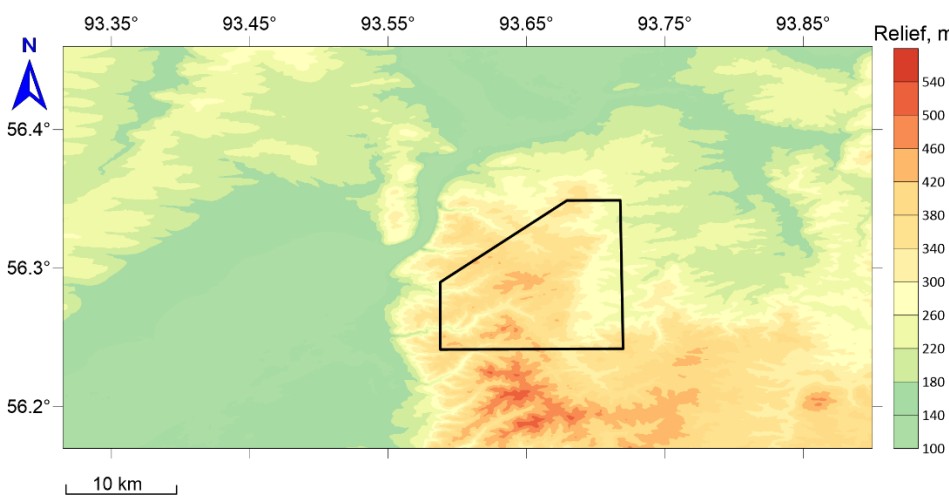

**Figure 1.** Original digital elevation model.

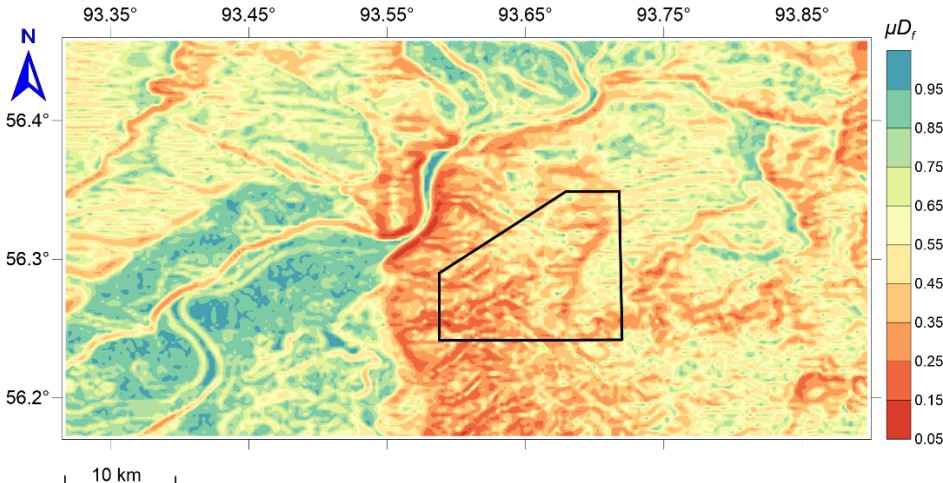

**Figure 2.** Measure of dynamic activity index of relief, $\mu L^1_{Re}(w)$.

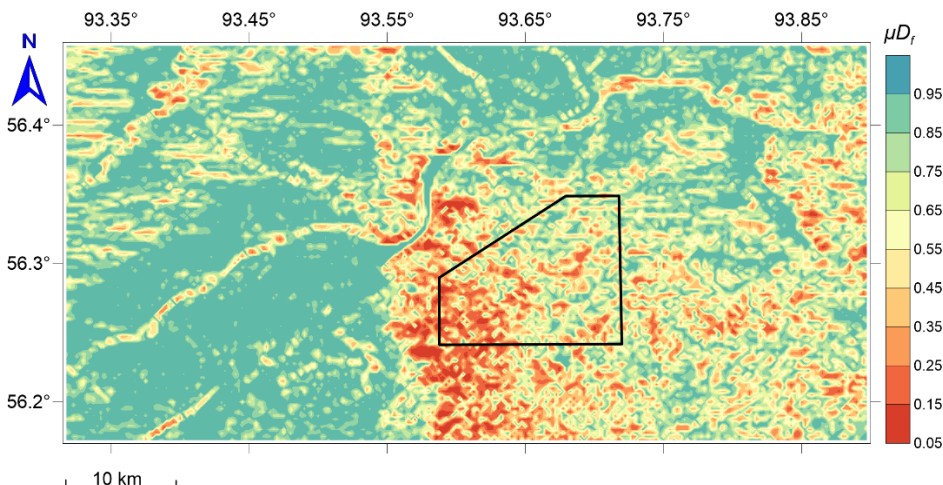

**Figure 3.** Measure of dynamic activity index of relief, $\mu L^2_{Re}(w)$.

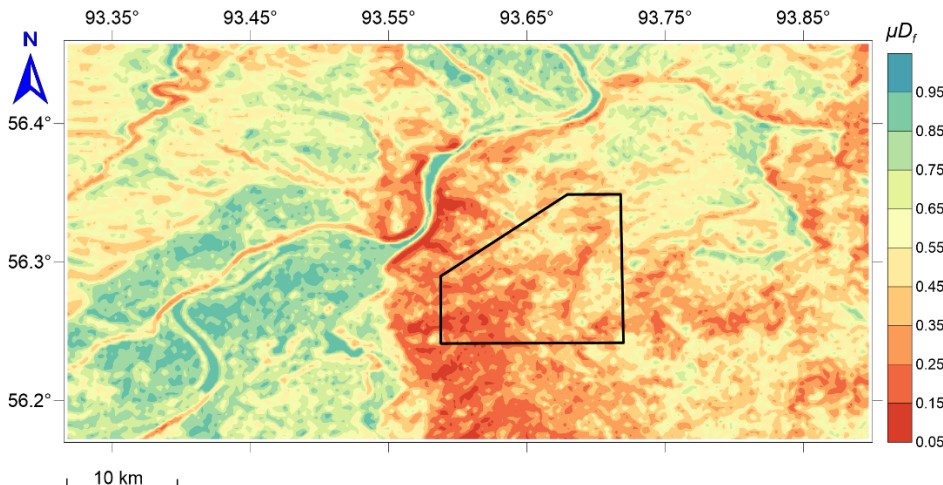

**Figure 4.** Measure of dynamic activity index of relief, $\mu \nabla_{Re}(w)$.

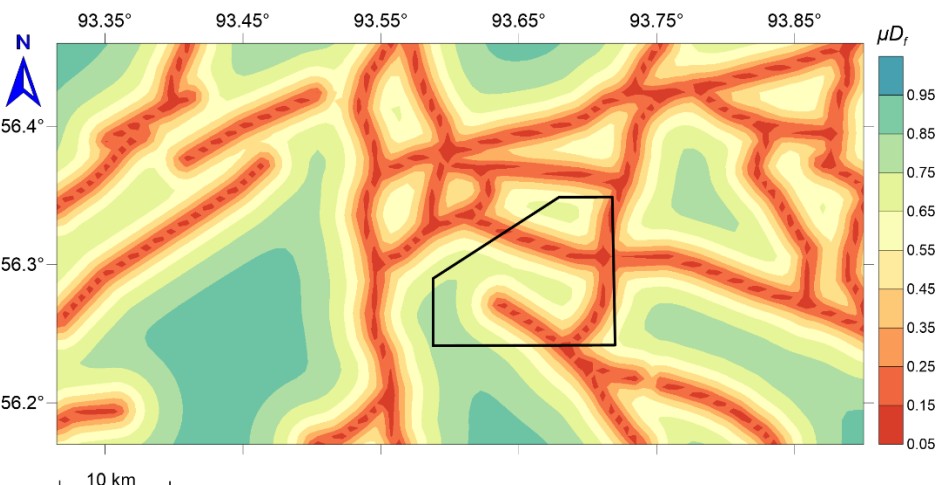

**Figure 5.** Measure of dynamic activity index of proximity to tectonic faults, $\mu^{\rho}(w, \mathcal{P})$.

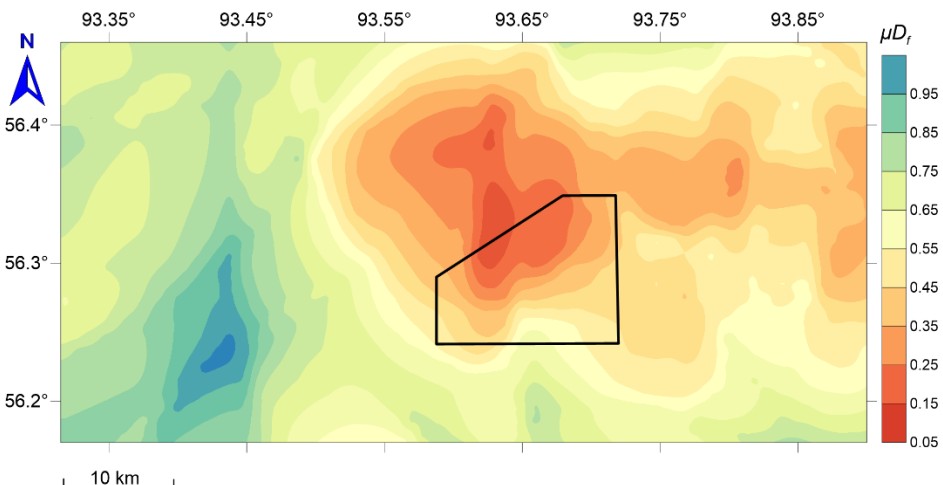

**Figure 6.** Measure of dynamic activity index of fault density, $\mu^p(w)$.

The next MDAI referred to the stress-strain state. The indexes for stress tensor components $\sigma_{xx}$, $\sigma_{yy}$ and stress intensity $\sigma_{int}$, which were obtained from the results of the numerical modeling in [30], were calculated. The indexes for deformation parameters of the Earth's crust, derived from GNSS measurements, were also calculated [19,41]. The components of the strain tensor $e_{xx}$, $e_{yy}$ and the dilatation strain $E_{Dil}$ were used as input data. The transformation of the stress-strain state parameters was carried out based on the conversion of these parameters to a gradient scale in the range 0 to 1. For this purpose, indicators $a$, $b$, $c$, and $d$ were determined. They correspond to the following measurements of the dynamic activity in Table 2: $a$, maximum value of stresses/strains; $b$, boundary value of stresses or strains on the positive part of the scale identified by expert judgment; $c$, boundary value of stresses or strains on the negative part of the scale identified by expert judgment; $d$, minimum value of stresses/strains. For each $x_i$ interval, a transformation was performed according to Equation (14). Thus, each interval of stress and strain values $x_i$ corresponds to a certain calculation formula and interval of the dynamic activity measure.

$$
x_i \in \begin{array}{c} [a;b] \\ \left[b; \frac{b+c}{2}\right] \\ \left[\frac{b+c}{2}; c\right] \\ [c;d] \end{array} \rightarrow \left\{ \begin{array}{l} 0.5 \times \frac{x-a}{b-a} \\ 0.5 + \frac{x-b}{c-b} \\ 1.5 - \frac{x-b}{c-b} \\ 0.5 \times \frac{d-x}{d-c} \end{array} \right. \tag{15}
$$

**Table 2.** Dynamic activity measure intervals based on stress and strain valuation.

| Intervals | Value |
|:---:|:---:|
| $[a;b]$ | 0–0.5 |
| $\left[b; \frac{b+c}{2}\right]$ | 0.5–1 |
| $\left[\frac{b+c}{2}; c\right]$ | 1–0.5 |
| $[c;d]$ | 0.5–0 |

Below are shown MDAIs of stress-strain state. Stress values $\sigma_{xx}$, $\sigma_{yy}$, and $\sigma_{int}$ are given in Figures 7–9 and strain values $e_{xx}$, $e_{yy}$, and $E_{Dil}$ in Figures 10–12.

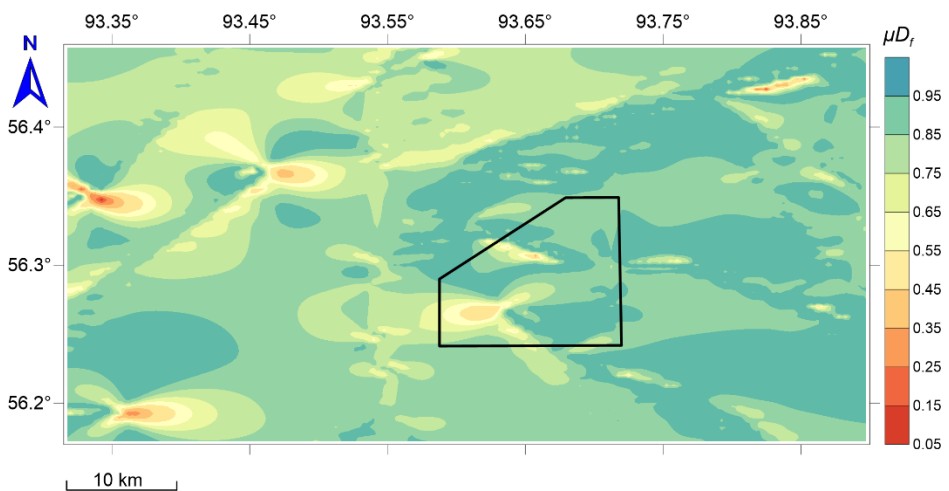

**Figure 7.** Measure of dynamic activity index of stress tensor component, $\mu\sigma_{xx}(w)$.

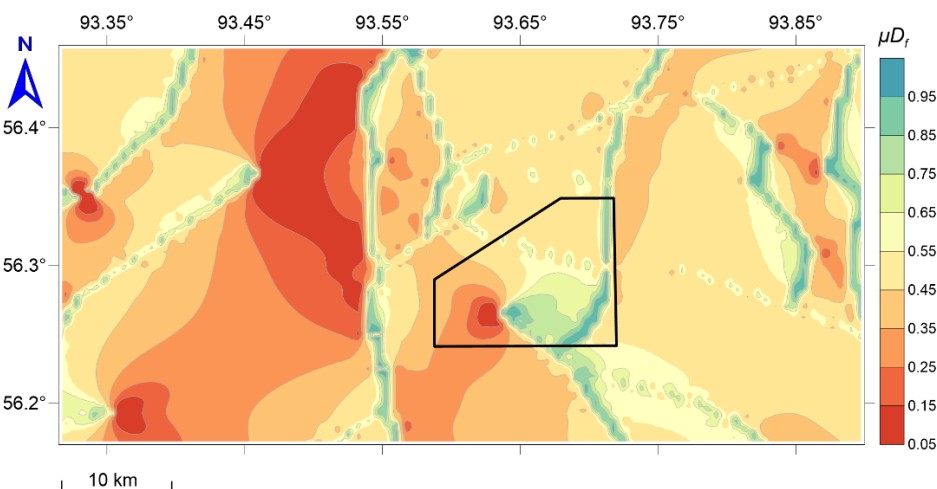

**Figure 8.** Measure of dynamic activity index of stress tensor component, $\mu\sigma_{yy}(w)$.

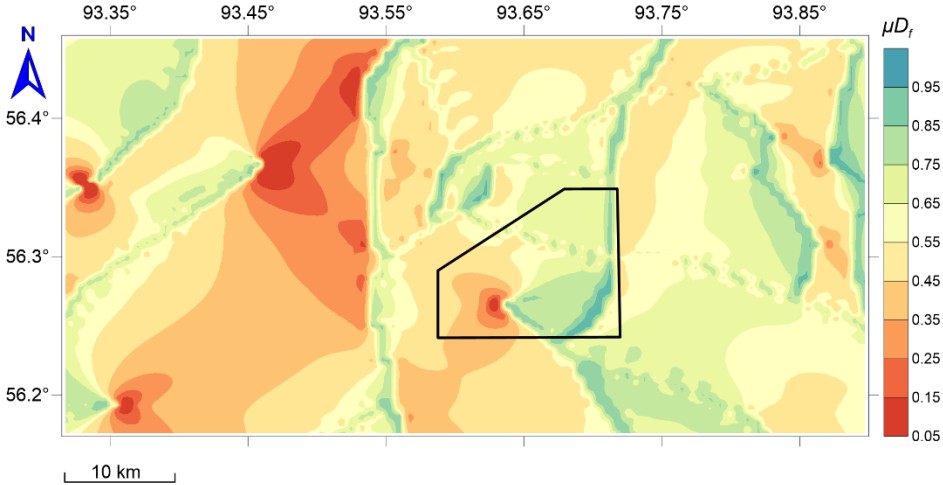

**Figure 9.** Measure of dynamic activity index of stress intensity, $\mu\sigma_{int}(w)$.

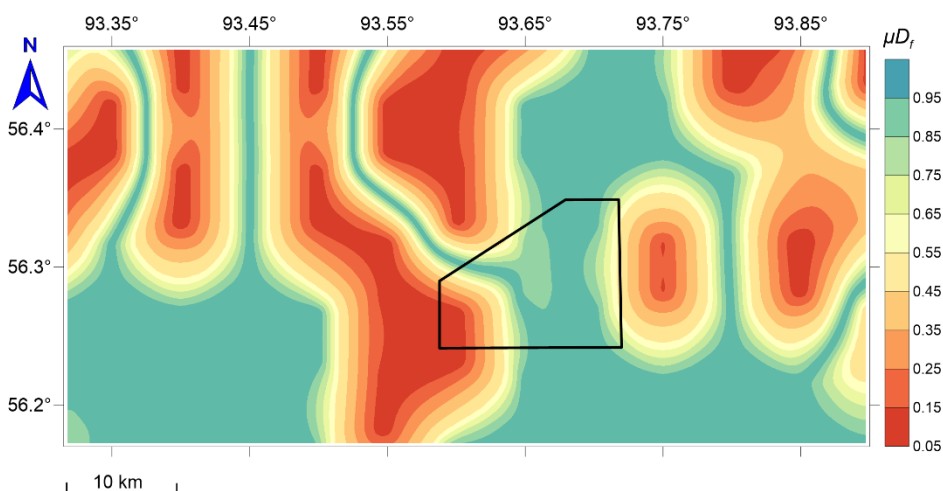

**Figure 10.** Measure of dynamic activity index of strain tensor component, $\mu e_{xx}(w)$.

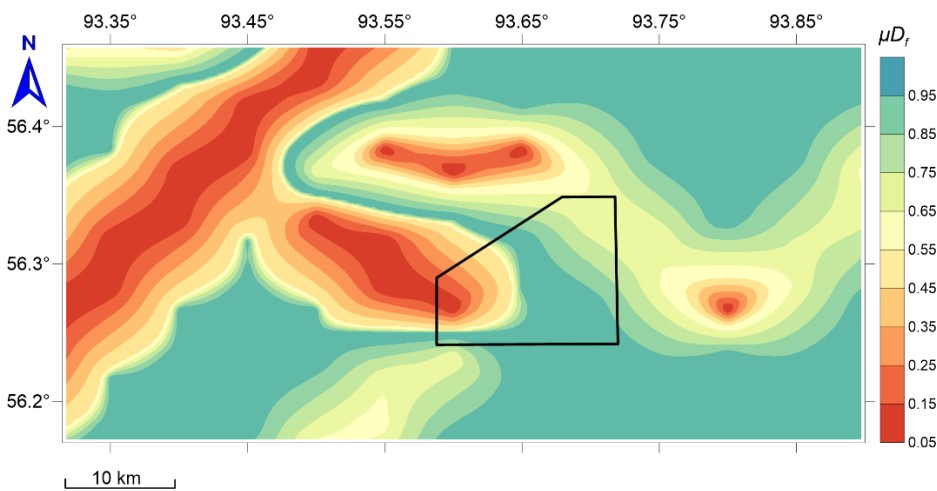

**Figure 11.** Measure of dynamic activity index of strain tensor component, $\mu e_{yy}(w)$.

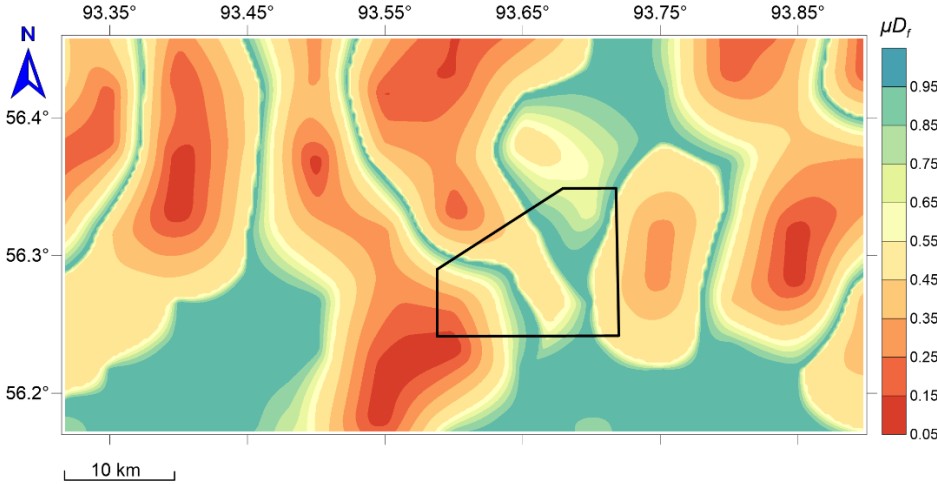

**Figure 12.** Measure of dynamic activity index of dilatation strain, $\mu E_{Dil}(w)$.

To determine the MDAI of the magnetic field, we used Equation (5) for magnetic field variability index $L^2_{Mag}(w)$ and Equation (8) for magnetic field variation gradient $\left|\nabla_{Mag}\right|(w)$.

Figure 13 shows distributions of integral measures of geodynamic safety and graphs of the distribution of their values by intervals within selected groups: geomorphological variability $S_{Re}(w)$, tectonic faults density $S_{\rho}(w)$, stress-strain state $S_{sss}(w)$, and magnetic field anomalies $S_{Mag}(w)$:

$$S_{Re}(w) = 1 - \frac{\mu L_{Re}^1(w) + \mu L_{Re}^2(w) + \mu \nabla_{Re}(w)}{3} \tag{16}$$

$$S_{\rho}(w) = 1 - \frac{\mu^{\rho}{}_{\rho}(w) + \mu^{\rho}(\Pi, \mathcal{P})}{2} \tag{17}$$

$$S_{sss}(w) = \frac{1}{2} \times \left( \frac{\mu\sigma_{xx}(w) + \mu\sigma_{yy}(w) + \mu\sigma_{int}(w)}{3} + \frac{\mu e_{xx}(w) + \mu e_{yy}(w) + \mu E_{Dil}(w)}{3} \right) \tag{18}$$

$$S_{Mag}(w) = 1 - \frac{\mu L_{Mag}^2(w) + \mu \nabla_{Mag}(w)}{2} \tag{19}$$

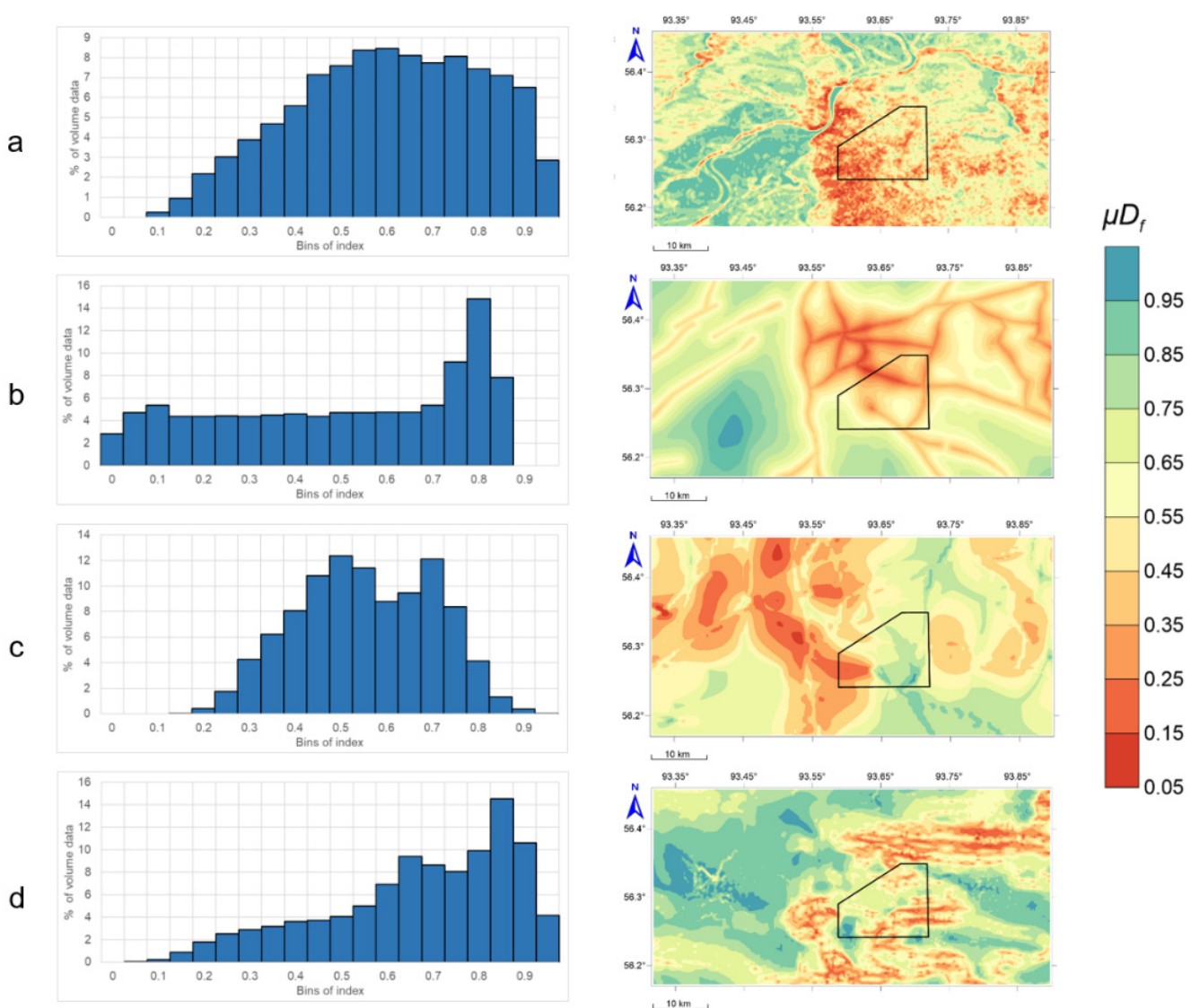

**Figure 13.** Integral measures geodynamic safety histograms values into groups: (**a**)—measure of safety by geomorphological variability, $S_{Re}(w)$; (**b**)—measure of safety by tectonic fault density, $S_{\rho}(w)$; (**c**)—measure of safety by stress-strain state, $S_{sss}(w)$; (**d**)—measure of safety by magnetic field anomalies, $S_{Mag}(w)$.

The distributions of the integral measures within the groups differ significantly. The integral measure of geomorphological variability $S_{Re}(w)$ is characterized by a uniform distribution of data in the measurement interval from 0.45 to 0.9 (Figure 13a). The geomorphological variability in the study area is not high. A large part of the area is flat with low elevation differences (Figure 1), but small areas of high-degree geomorphological variability can also be observed with a relative geodynamic instability measurement in the range of 0.1–0.25. The integral measure of tectonic fault density $S_\rho(w)$ has a uniform distribution of values between 0 and 0.75. About 31% of all data fall within the relative geodynamic stability interval of 0.75–0.9 (Figure 13b). The integral measure of the stress-strain state $S_{sss}(w)$ has a distribution close to normal, and about 85% of all data fall into the 0.25–0.75 transition range (Figure 13c). Up to 14% of all nodes are in the area of relative geodynamic stability. The integral measure of magnetic field anomaly variability $S_{Mag}(w)$ is not uniformly distributed over the intervals. The 0.75–1 interval of relative geodynamic stability accounts for up to 47% of all data, the 0.25–0.75 interval of transient values for about 50% of all data, and the 0–0.25 interval of relative geodynamic instability for about 3%.

## 4. Discussion

Correlation matrices and correlation strength thresholds were calculated for dynamic indexes $D_f$ and their MDAI $\mu D_f$. Pearson's correlation coefficient is presented in the matrix:

$$r = \frac{\sum((x_i - \bar{x}) \times (y_i - \bar{y}))}{\sqrt{\sum\left((x_i - \bar{x})^2 \times \sum(y_i - \bar{y})^2\right)}} \tag{20}$$

The lower threshold for the presence of correlation was determined using Student's *t*-test Equation (21) and the intervals for the strength of the correlation according to Equation (22):

$$r_0 = \frac{t}{\sqrt{t^2 + n - 2}} \tag{21}$$

$$r_{int} = \frac{1 - r_0}{3} \tag{22}$$

Correlation strength intervals were determined for the dataset used (27,495 rows in each of the indicators and a significance level of 0.95): weak correlation in the 0.012–0.341 interval, medium correlation in the 0.341–0.671 interval, and strong correlation in the 0.671–1 interval. The strength of the correlation was represented as a discrete color scale.

Figure 14 shows the correlation matrix of MDAI $\mu D_f$. As shown in the figure, 65 out of 78 correlation values of MDAI $\mu D_f$ have a weak correlation relationship. This is a good indicator in terms of data analysis as the features must be noncollinear, otherwise, the aggregation ability of the integral measure of geodynamic safety is reduced due to the high dispersion of the data $S_F(w)$. Medium and strong correlations are found within index group $\mu D_f$. In the relief group, indexes of geomorphological variability $\mu L_{Re}^1(w)$ and $\mu L_{Re}^2(w)$ have a strong correlation with the index of relief gradient $\mu\nabla_{Re}(w)$. MDAIs $\mu L_{Re}^1(w)$ and $\mu L_{Re}^2(w)$ have a medium correlation. The measures of proximity to tectonic faults index $\mu\rho(w,\mathcal{P})$ and fault density $\mu\rho(w)$ have a strong correlation, as the indices show high spatial correspondence [42,45] (Figures 5 and 6). The fault group measurements have an average correlation strength with the stress measurements $\mu\sigma_{yy}(w)$ and $\mu\sigma_{int}(w)$. This is primarily due to the mathematical model for calculating the stress state [43], which considers the fault tectonics of the area. In the stress group, $\mu\sigma_{yy}(w)$ and $\mu\sigma_{int}(w)$ have a strong correlation relationship, as the stress $\sigma_{yy}$ has the greatest contribution to the stress intensity $\sigma_{int}$ [44]. The MDAIs of the strain tensor components $\mu e_{xx}(w)$ and $\mu e_{yy}(w)$, determined from GNSS observations [19], have a weak correlation. The MDAI of dilatation strain $\mu E_{Dil}(w)$ has a strong correlation with the measure $\mu e_{xx}(w)$ and a medium correlation with the measure $\mu e_{yy}(w)$; therefore, their spatial relationship can be seen (Figures 7–9). This is explained by the formula for calculating strain dilatation, which includes the components of the

strain tensor $e_{xx}$ and $e_{yy}$ [41]. The average correlation strength was found between the measure of magnetic field anomaly variability $\mu L^2_{Mag}(w)$ and the gradient of magnetic field anomaly variability $\mu \nabla_{Mag}(w)$. Internal correlation of feature groups was due to either interdependent formulas for deriving the initial properties of the dynamic indicators $D_f$, or the same set of initial feature data F (as in the case of topography or magnetic field anomalies). Table 3 shows the correlation of the integral geodynamic safety groups: safety measure of geomorphological variability, $S_{Re}(w)$; safety measure of tectonic faults density, $S_\rho(w)$; safety measure of a stress-strain state, $S_{sss}(w)$; and safety measure of magnetic field anomalies, $S_{Mag}(w)$. Their correlation everywhere indicates a weak correlation or no correlation at all. This characterizes these indexes as reflecting different properties of the geological environment and as independent datasets.

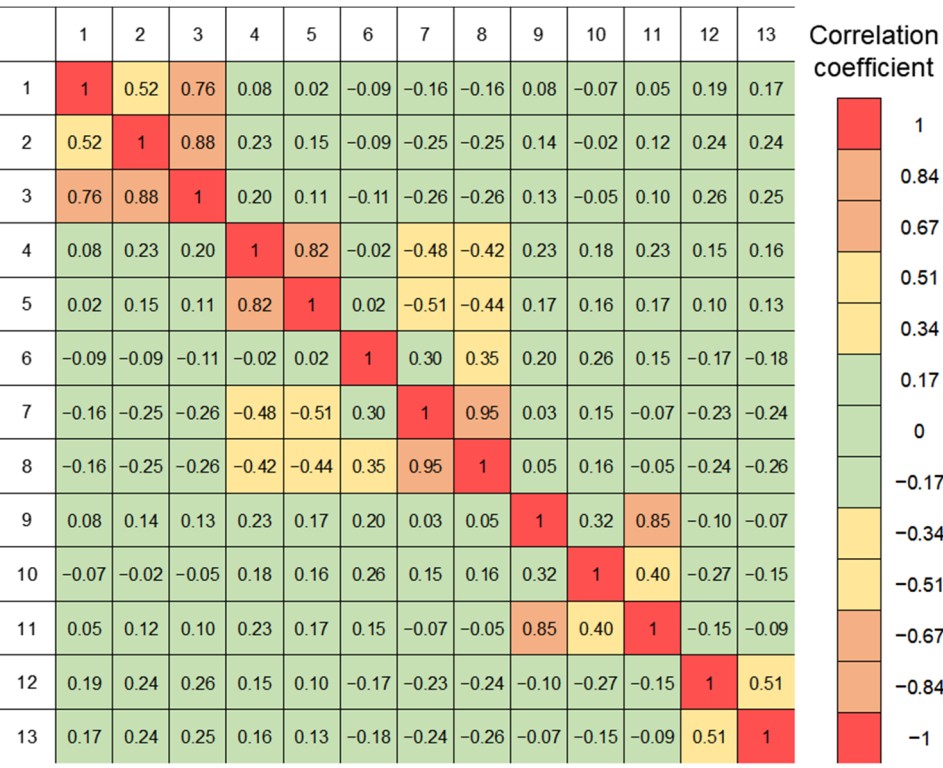

**Figure 14.** Correlation matrix the measures of dynamic activity indexes $\mu D_f$: 1—relief index, $\mu L^1_{Re}(w)$; 2—relief index, $\mu L^2_{Re}(w)$; 3—relief index, $\mu \nabla_{Re}(w)$; 4—proximity to tectonic faults index, $\mu^\rho(w, \mathcal{P})$; 5—fault density index, $\mu^\rho(w)$; 6—index of stress tensor component $\mu \sigma_{xx}(w)$; 7—index of stress tensor component, $\mu \sigma_{yy}(w)$; 8—index of stress intensity, $\mu \sigma_{int}(w)$; 9—index of strain tensor component, $\mu e_{xx}(w)$; 10—index of strain tensor component, $\mu e_{yy}(w)$; 11—index of dilatation strain $\mu E_{Dil}(w)$; 12—magnetic field variability index, $\mu L^2_{Mag}(w)$; 13—magnetic field variation gradient, $\mu \nabla_{Mag}(w)$.

**Table 3.** Correlation matrix of integral measures of geodynamic safety.

|  | $S_{Re}(w)$ | $S_\rho(w)$ | $S_{sss}(w)$ | $S_{Mag}(w)$ |
|---|---|---|---|---|
| $S_{Re}(w)$ | 1 | 0.13 | −0.03 | 0.28 |
| $S_\rho(w)$ | 0.13 | 1 | 0.04 | 0.16 |
| $S_{sss}(w)$ | −0.03 | 0.04 | 1 | −0.28 |
| $S_{Mag}(w)$ | 0.28 | 0.16 | −0.28 | 1 |

The integral measure of geodynamic safety $S_F(w)$ was calculated based on a combination of four integral measures for relief, faults, stress-strain state, and magnetic field anomalies:

$$S_F(w) = \frac{S_{Re}(w) + S_\rho(w) + S_{sss}(w) + S_{Mag}(w)}{4} \tag{23}$$

The spatial distribution of the integral measure of geodynamic safety $S_F(w)$ is shown in Figure 15, and the distribution of measurements by intervals is shown in Figure 16. Green corresponds to the most stable state, red to the least stable, and yellow to the intermediate zones.

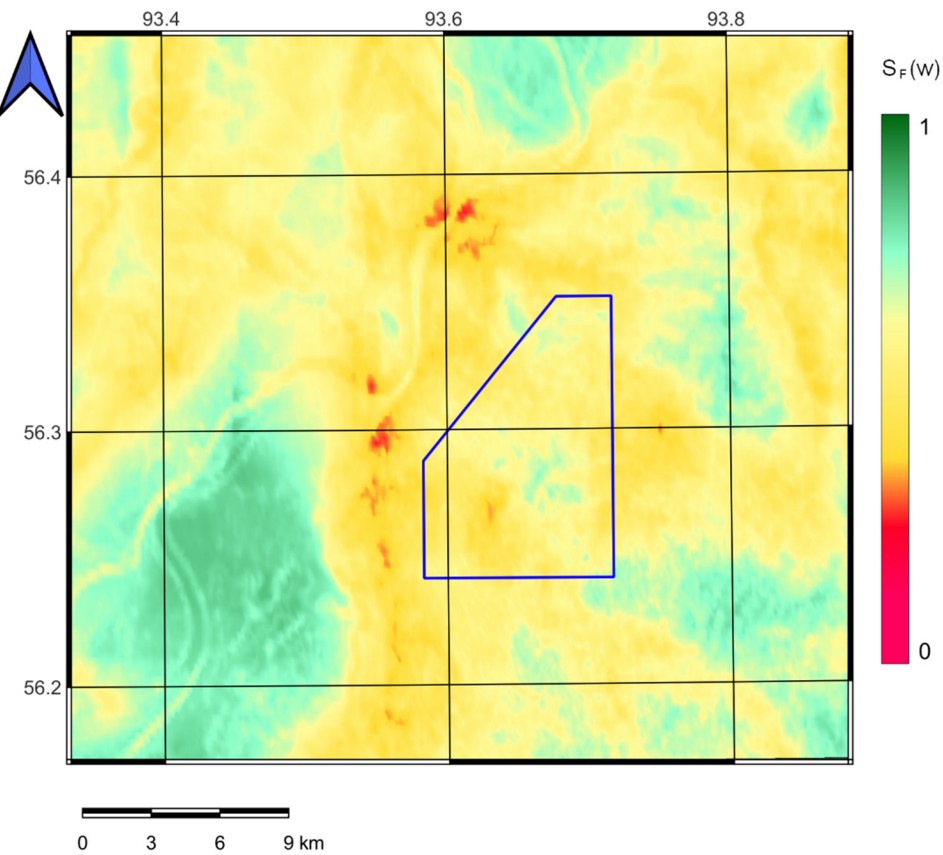

**Figure 15.** Map of the integral measure of geodynamic safety, $S_F(w)$; blue contour is boundary of HLRW site isolation.

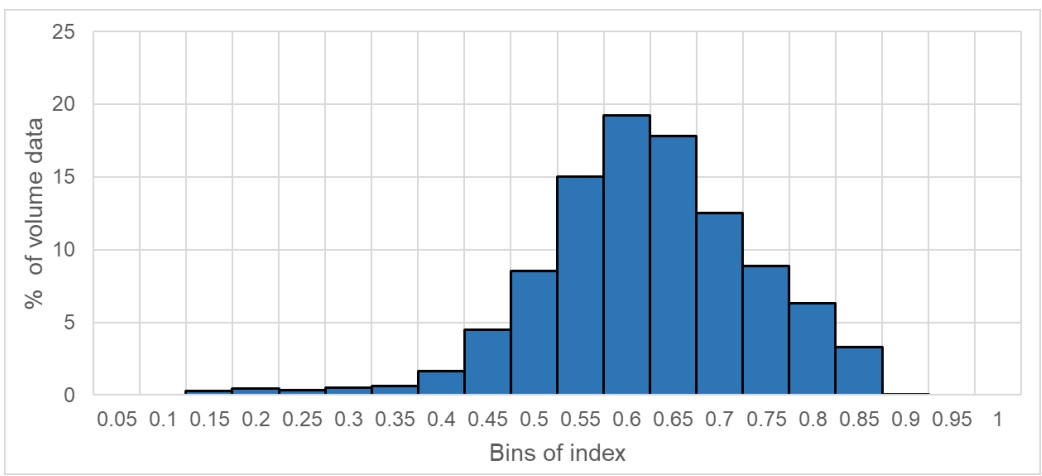

**Figure 16.** Histogram of the integral measure of geodynamic safety, $S_F(w)$.

According to the results of the integral measure ranking (Table 1), nodes of relative geodynamic instability $S_F(w) \leq 0.25$ account for an area of about 14 km$^2$ (305 nodes).

Most of these nodes belong to the area of interaction between major tectonic structures and sharp variations in height gradient, which are actively expressed in abnormal values of all measures of activity of dynamic indexes $\mu D_f$. The range of hazardous nodes is very small and occupies about 1.1% of the total volume. Nodes of geodynamic stability $S_F(w) \geq 0.75$ correspond to flat areas of relief located in the centers of main structural-tectonic blocks, equidistant from tectonic faults and outside zones of anomaly stress-strain state. Nodes of relative geodynamic stability occupy an area of about 9.6%. Transition zones $S_F(w) \in (0.25; 0.75)$ are not uniformly distributed. Most of the transition interval nodes are associated with areas of local tectonic interaction, areas of moderate terrain variability, and stress state within background values. The transition interval nodes occupy the largest study area, about 89.3%.

## 5. Conclusions

Assessment of the stability of the geological environment during disposal HLRW for the period of their radiobiological hazard (more than ten thousand years) is a very difficult task in the field of Earth sciences. Currently, this complex problem is being solved in several countries. For example, in Sweden («Forsmark») and Finland («Aspo»), similar facilities are already under construction. The proposed methodology for assessing sustainability complements the methods used in these and other countries to assess the suitability of the geological environment for the disposal of HLRW. The degree of stability of the environment as a geodynamic active system is related to the distribution features of the geospatial data complex, including the characteristics of the relief, tectonic faults, geophysical fields, stresses, etc. The search for anomalies and morphological patterns in the distribution of data allows us to identify possible sites of the destruction of structural blocks.

The presented methodology for assessing the stability of the geological environment based on the methods of DMA and fuzzy logic is informed by the analysis of dynamic indicators as functions of geodynamic activity of the environment by groups: geological and geophysical data, geomorphological variability of relief, the density of tectonic faults, stress-strain state, and magnetic and gravity field anomalies for the area [46–49]. A detailed analysis of the correlation matrix of the MDAI for four groups' characteristics of the environment $\mu D_f$ demonstrated that the absolute majority have a weak correlation. This is a positive point, as the features should not be collinear. Otherwise, due to the high variance of the data, the generalizing ability of the integral measure of geodynamic safety $S_F(w)$ would decrease. Medium and strong correlations were found within the groups of indicators $\mu D_f$.

The integral measure of geodynamic safety $S_F(w)$, which combines different MDAIs, allows geodynamic zoning of the HLRW disposal area and specifying places for field instrumental observations.

The effectiveness of the presented methodology depends on the quality and volume of the source data. In our study, the methodology only demonstrates a possible approach to the systemic assessment of stability, and it could be expanded in the future both in terms of increasing features and by creating new algorithms. It could also be used for the identification of linear anomalies associated with hidden tectonic faults and dangerous zones of geodynamic instability.

It was found that most of the studied territory, where HLRW disposal is currently planned, is located in an intermediate zone in terms of the degree of geodynamic stability $S_F \geq 0.75$. The number of dangerous clusters $S_F \leq 0.25$ is very small and occupies about 1.1% of the entire territory.

**Author Contributions:** Conceptualization, S.M.A. and V.N.T.; methodology, S.M.A. and I.V.L.; software, I.O.B.; validation, I.V.L., I.O.B. and M.A.P.; formal analysis, A.I.M.; investigation, M.A.P.; resources, I.V.L. and A.I.M.; data curation, I.V.L.; writing—original draft preparation, S.M.A., I.V.L. and A.I.M.; writing—review and editing, V.N.T., M.A.P., I.O.B. and I.V.L.; visualization, I.V.L. and A.I.M.; supervision, V.N.T.; project administration, S.M.A.; funding acquisition, V.N.T. All authors have read and agreed to the published version of the manuscript.

**Funding:** This research was funded by the Russian Science Foundation (project No. 18-17-00241).

**Institutional Review Board Statement:** Not applicable.

**Informed Consent Statement:** Informed consent was obtained from all subjects involved in the study.

**Data Availability Statement:** The data presented in this study are available upon request from the corresponding author. The data are temporarily not publicly available due to research policies and implementation of research programs.

**Acknowledgments:** This work employed data provided by the Shared Research Facility «Analytical Geomagnetic Data Center» of the Geophysical Center of RAS (http://ckp.gcras.ru/ (accessed on 27 January 2022)).

**Conflicts of Interest:** The authors declare no conflict of interest.

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
