# Peer review of "Dynamic Activity Index for Feature Engineering of Geodynamic Data for Safe Underground Isolation of High-Level Radioactive Waste"

_applsci, doi:10.3390/app12042010_

Round 1
Reviewer 1 Report
Overview and overall recommendation:
This manuscript is focused on the development of a new approach for feature engineering in geosciences based on the implementation of measured dynamic activity index (MDAI). The authors applied this new approach to assess the geological stability conditions of a high-level radioactive waste repository (HLW). The geomorphological variability of relief, the density of tectonic faults, the stress-strain state and the magnetic field anomalies of the area have been analyzed. A detailed analysis of the correlation matrix of MDAI for the four groups characteristics demonstrated that the absolute majority have a weak correlation.
The manuscript follows the article structure suggested by Applied Sciences. It contains interesting material for the scientific community working in the field of featured engineering. However, I believe that it needs to be revised and rewritten in some parts to be published.
The section which needs a major revision is the Introduction due to the lack of: (1) Some of the most updated references in the field of feature engineering; (2) An overview of the published results of the assessment and prediction of the geodynamic stability of the geological environment in deep geological repositories of HLW (including references) such as the proposed in the Nizhnekansky massif in Russia and others; (3) The general description of the deep geological repository (DGR) system in granite and in particular of the place selected in the paper; and (4) A short paragraph at the end of the Introduction section mentioning the different sections of the paper.
As in the title of the paper appears the term “high-level radioactive waste” and now there is no information in the paper about this type of disposal concept, I strongly believe that it is necessary to include in the introduction (or in a new section) a general description of the DGR system in granite including the different barriers (vitrified glass and/or spent fuel, metallic canister, bentonite buffer, granite host rock), depth, current status in the world, … Moreover, there is no information in the paper about the selected HLW disposal concept in granite in which the geological stability conditions were studied applying the new approach presented here. This information should be also included in the paper.
My recommendation for this article is minor revision. In order to be accepted for publication a list of comments is provided to improve the quality of the manuscript.
Reviewer’s comments:
Title and authors
Page 1, lines 7-9: I suggest to add the full address of the 3 institutions or at least the location and country.
Abstract
Page 1, line 20: In the field of the radioactive waste management the used term for high-level radioactive waste is “HLW”. This term refers only to the vitrified nuclear waste. This means that the fuel elements are unloaded from the reactor of a nuclear power plant and are kept releasing heat and radioactivity to a level at which they may be converted into vitrified waste and placed at the HLW disposal cell. There is also another term for the spent fuel (SF). In that case the fuel elements are not converted into vitrified waste and will be placed after a cooling time in a SF disposal cell. Depending on the radioactive waste management of each country there will be HLW or SF or both. I am not sure about which of one is focused this paper. Therefore, I suggest to change the term “HLRW” in the abstract and in the whole paper by a more general “HLW and SF”.
1 Introduction
Page 1, line 24. In my opinion the main and the only weakness of the article is that the introduction does not provide sufficient background and the most updated relevant references. In order to solve this problem, I suggest the following actions:
1) Improve and extent the explanation of feature engineering presented in the first paragraph and include some of the most updated references in this field.
2) Include an overview of the available published results of the assessment and prediction of the geodynamic stability of the geological environment in deep geological repositories of HLW (including references) such as the proposed in the Nizhnekansky massif in Russia and others. The publication of Tatarinov et al. (2021), [15] in the paper, could be an example.
3) Add a short general description of the deep geological repository (DGR) system in granite including the different barriers (vitrified glass and/or spent fuel, metallic canister, bentonite buffer, granite host rock). After that, add also information about the place selected in the paper to apply the new approach and assess its geological stability conditions. It is not clear for me what is the selected DGR of HLW. In my opinion, a minimal information about the DGR should be provided to any potential reader of the paper. Now at the beginning of the second paragraph the HLW is mentioned but no more explanation is added. An example of this could be found in the introduction of Morozov et al. (2012) for Nizhnekanskii granitoid massif ([12] in the paper).
4) Add a few lines at the end of the Introduction section mentioning the different sections of the paper.
2 Methodology for assessing the integral index of dynamic stability of a structural 54 tectonic block
Page 2, lines 56-60. I suggest to rewrite these lines, specially the first two lines (“Below is information about the methodology that was developed by the Geophysical Center RAS.”). I mean that a little information about the development of this methodology should be added in the paper (including references).
Page 2, lines 64-65. Please, provide a little information about the studied HLW disposal cell (location, depth, properties of the granite, …)
3 Feature engineering based on of dynamic stability index
Page 3, line 113. I propose to delete the word “of” in the section 3 title and use “the”.
Page 3, line 114. According to the journal style, the section 3.1 title should be on the left side (similar to the rest of the titles section). I also propose to change the title to “Needed data”.
Page 3, line 132. The same for section 3.1 title (Feature calculation). This section must be 3.2 and not 3.1.
Page 5, lines 162-163. The first letter of the word “figure” of Figures 5 and 6 in the text of the paper should be in capital letters (Figure).
Page 7, line 189. The word “formula” of formula 14 in the text of the paper should be replaced by “Equation (14)”.
Page 7, line 195. Same comment for “figures 7, 8 and 9” than in comment of page 5, lines 162-163.
Page 9, line 208. Same comment for “formula 5” than in comment of page 7, line 189.
4 Discussion
Page 11, line 240. Same comment for “formula 21 and 22” (twice) than in comment of page 7, line 189.
Page 12, line 291. Same comment for “figure 15 and 16” (twice) than in comment of page 5, lines 162-163.
Page 13, line 310. Add a full stop at the end of the figure caption.
5 Conclusions
Page 14, lines 314-316. The following lines “Assessment of the stability geological environment during disposal HLRW for the period of their radiobiological hazardous (more than 10 thousand years) is a very difficult task in the field of Earth sciences. It is currently not solved in any country in the world.” should be rewritten. In my opinion it should be mentioned that the long-term (more than 10.000 years) geological stability of the host rock barrier (granite) of a HLW or SF deep geological repository is a difficult challenge. Nowadays, there is a DGR in granite almost in construction phase in Finland and other countries (Sweden, France, …) are in advances stages of the approval phase. These countries (and others) have studied the geological stability of the host rock using different approaches. Although perhaps the approach presented in this paper has not been used. Therefore, the statement about that the assessment of the geological stability of a DGR is not solved in any country in the world is not true.
References
As I have mentioned previously in the comment of Page 1, line 24, some of the most updated references in the field of feature engineering and about the published results of the assessment and prediction of the geodynamic stability of the geological environment in DGR of HLW should be added to the paper.
Author Response
Dear Reviewer,
Thank you so much for reading our article. We have tried to take into account most of your comments. You can find them in the attached file.

Reviewer 2 Report
The presented research issue on the dynamic activity index seems to be interesting from the point of view of better understanding of dynamic phenomena occurring in the rock mass. Of particular interest is the geodynamic coefficient, which can be useful in estimating the risk of damage. Below are some comments and suggestions:
- In the abstract, line 20, mention should be made of the abbreviation HLRW.
- In the introduction, one of the methods of monitoring the increase in stresses in the rock mass is the use of string sensors, which are particularly important for the rock mass, which is prone to relief seismic energy in the form of tremors (/doi.org/10.3390/en13112998).
- Definitions of letters on lines 184-187 should be placed under expresion 2; this would significantly improve comprehension and readability.
- Subchapters 3.1 and 3.2 have the same numbering, they should be corrected. In addition, subsection 3.1 is mainly based on literature and is too short; I propose to combine it with another subsection.
- In the chapter 3.2 a few sentences should be written about the fault parameters, in particular: throw, amplitude, fault plane angle, horizontal range. How do fault parameters affect the dynamic activity index?
- In the fourth chapter about the discussion, reference should be made to several literaturę items where the geodynamic coefficient of the fault regions is determined.
- In the summary, one conclusion should be written regarding the numerical values of the geodynamic coefficient.
Author Response

(The authors gave the same response as above.)
